# Exploring the feasibility of technological visuo-cognitive training in Parkinson's: Study protocol for a pilot randomised controlled trial

Julia Das[1,2], Rosie Morris[1,2], Gill Barry[1], Rodrigo Vitorio[1], Paul Oman[3], Claire McDonald[4], Richard Walker[2], Samuel Stuart[1,2]*

1 Department of Sport, Exercise & Rehabilitation, Northumbria University, Newcastle upon Tyne, United Kingdom, 2 Northumbria Healthcare NHS Foundation Trust, North Tyneside General Hospital, North Shields, United Kingdom, 3 Department of Mathematics, Physics & Electrical Engineering, Northumbria University, Newcastle upon Tyne, United Kingdom, 4 Gateshead Health NHS Foundation Trust, Gateshead, United Kingdom

* sam.stuart@northumbria.ac.uk

**Data Availability Statement:** No datasets were generated or analysed during the current study. All

## Abstract

Visual and cognitive dysfunction are common in Parkinson's disease and relate to balance and gait impairment, as well as increased falls risk and reduced quality of life. Vision and cognition are interrelated (termed visuo-cognition) which makes intervention complex in people with Parkinson's (PwP). Non-pharmacological interventions for visuo-cognitive deficits are possible with modern technology, such as combined mobile applications and stroboscopic glasses, but evidence for their effectiveness in PwP is lacking. We aim to investigate whether technological visuo-cognitive training (TVT) can improve visuo-cognitive function in PwP. We will use a parallel group randomised controlled trial to evaluate the feasibility and acceptability of TVT versus standard care in PwP. Forty PwP who meet our inclusion criteria will be randomly assigned to one of two visuo-cognitive training interventions. Both interventions will be carried out by a qualified physiotherapist in participants own homes (1-hour sessions, twice a week, for 4 weeks). Outcome measures will be assessed on anti-parkinsonian medication at baseline and at the end of the 4-week intervention. Feasibility of the TVT intervention will be assessed in relation to safety and acceptability of the technological intervention, compliance and adherence to the intervention and usability of equipment in participants homes. Additionally, semi structured interviews will be conducted to explore participants' experience of the technology. Exploratory efficacy outcomes will include change in visual attention measured using the Trail Making Test as well as changes in balance, gait, quality of life, fear of falling and levels of activity. This pilot study will focus on the feasibility and acceptability of TVT in PwP and provide preliminary data to support the design of a larger, multi-centre randomised controlled trial. This trial is registered at isrctn.com (ISRCTN46164906).

relevant data from this study will be made available upon study completion.

**Funding:** This work forms part of PhD study being undertaken by JD and has been funded by a Northumbria University PhD studentship in collaboration with Senaptec Inc. (Beaverton, Oregon, USA) (PI: Dr Samuel Stuart). Dr Stuart is supported, in part, by a Parkinson's Foundation Post-doctoral Fellowship for Basic Scientists (PF-FBS-1898-18-21) and a Clinical Research Award (PF-CRA-2073).

**Competing interests:** The authors have declared that no competing interests exist.

# Introduction

Parkinson's disease is a common neurodegenerative disorder characterised by motor and non-motor symptoms (NMS) [1], which impact quality of life and increase carer burden [2, 3]. NMS are difficult to treat and include autonomic dysfunction, pain, cognitive dysfunction and sensory deficits including visual dysfunction [4]. Visual and cognitive impairments are common in people with Parkinson's (PwP), with up to 80% of people reporting at least one visual symptom [5–8] and 40% showing mild cognitive impairment at diagnosis [9]. Visual impairments range from retinal changes (due to reduced dopamine) to more complex visuo-cognitive processing in higher order brain regions [4, 5, 8, 10–12]. Ophthalmology referral can treat retinal-related problems, such as impaired visual acuity and contrast sensitivity, with the application of corrective eye wear. However, interventions for deficits in higher order visual processes (spatial orientation, visual attention etc.) are lacking, as these are much more difficult to assess and treat [13]. Complexity arises due to visual and cognitive function being inter-related (termed visuo-cognition) and difficult to tease apart [12].

Visuo-cognitive impairments have been associated with motor deficits in PwP, specifically postural instability and gait impairment [7, 14–17]. This is further complicated by the fact that deficits in the ability to use proprioceptive feedback in PwP result in greater reliance on impaired visuo-cognitive function to carry out motor tasks [7, 18–22]. The combination of visuo-cognitive deficits and increased dependence on faulty visuo-cognitive information has a significant impact on motor symptoms in PwP, particularly gait and balance [8, 22, 23]. Specifically, visuo-cognitive impairments reduce the ability to compensate for underlying motor dysfunction in Parkinson's, which can lead to a decline in daily activities, increased risk of falls and reduced quality of life) [3, 24, 25].

There is a lack of pharmacological treatment options for visuo-cognitive deficits [26]. Therefore, demand for tailored visuo-cognitive rehabilitation strategies has increased which has led to development of studies examining specific intervention protocols [27]. For example, Camacho et. al developed an experimental protocol in 2019 looking at eye-movement training to improve voluntary saccade function in PwP [28]. Previous studies have shown some improvement in visuo-cognitive function in PwP using clinical interventions, such as eye movement training. For example, Zampieri et. al. in 2008 and 2009 suggested that eye movement exercises may improve gaze control in PwP, with carry over improvement in gait if combined with balance exercises [29, 30]. Similarly, a more recent study demonstrated a significant improvement in convergence insufficiency following two months of eye-movement training (vergence therapy) in a small cohort of PwP [31] While the findings of previous studies have contributed to the growing interest in the effects of eye-movement training in PwP [32–34], the treatments described in these trials often involve participants having to attend multiple training sessions per week at a laboratory/clinic/hospital or performing repetitive activities, such as pencil push ups several times per day [31]. Previous studies were limited by small sample sizes, strict inclusion criteria, and the burdensome protocols that resulted in significant drop-out rates and low participant uptake, which all limit the generalisability to clinical practice.

An alternative to eye-movement training is visuo-cognitive training programmes that are carried out online, mobile or on computer applications (apps) [35–38]. Visuo-cognitive training using technological devices have shown some efficacy in neurological populations [39–43]. Leung et al. (2015) carried out a meta-analysis to quantify the effects of visuo-cognitive training specifically in PwP [44]. Of the 7 studies included in their review, 6 involved some form of technology-based training [45–50], and improvements were seen in working memory, executive functioning, and processing speed, which are typically impaired in PwP [51]. A recent study by Shalmoni & Kalron (2020) was the first to investigate the use of specialist visuo-cognitive

technologies (such as stroboscopic / piezoelectric glasses) within a neurodegenerative population [52]. Specifically, they examined the effect of stroboscopic eyewear on visuo-cognitive function, gait, and static balance performance in people with multiple sclerosis. Their findings demonstrated that stroboscopic eyewear enhanced information processing speed immediately after training. While these findings were based on a multiple sclerosis population, the authors suggest that their research could have implications for the elderly and other neurological populations. Stroboscopic eyewear has primarily been used in healthy young adult populations to improve visuo-cognitive function [53, 54]; specifically anticipatory (visual processing speed / memory) and visual attentional reaction time [55, 56], visual acuity [57], hand-eye co-ordination [58] and information encoding [59]. Stroboscopic eyewear involves the intermittent reduction in visual input to create suboptimal visual conditions [55, 60], which can be used during basic tasks (e.g., throwing and catching) or combined with mobile training applications (e.g., screen-based reaction time tasks) [61]. The concept is based upon the premise that stroboscopic interruption of vision might enhance visuo-cognitive and motor control by reducing reliance on visuo-cognitive feedback loops and encouraging sensory re-weighting involving the use of other senses (i.e., proprioception, vestibular) within motor control [54, 62].

The use of a multimodal visuo-cognitive training intervention with the potential to target visual, cognitive and motor abilities has yet to be examined in PwP [63]. Therefore, research is required to determine the benefits of technological visuo-cognitive training (TVT) in PwP, as it may have potential as a novel non-pharmacological intervention for visuo-cognitive dysfunction. This manuscript describes the protocol that will be used to explore the feasibility of the TVT intervention as a potential rehabilitation tool in PwP by (1) evaluating the study design and procedures, (2) analysing the acceptability of the intervention by participants and (3) exploring efficacy of TVT to inform future trials. The interventions in this study have potential to address visuo-cognitive problems, balance and falls which are aligned with Parkinson's UK top ten research priority areas [64]. Findings from this study will inform the development of a future large-scale study to evaluate the impact of TVT on visuo-cognition in PwP.

## Study objectives

The primary aim is to investigate the feasibility and acceptability of a home-based technological visuo-cognitive training intervention in a sample of 40 PwP. We hypothesise that TVT will be feasible and acceptable to participants due to the novelty of the intervention, and the one-to-one support offered by the research Physiotherapist in participant homes will reduce the likelihood of study drop-outs. The secondary aim is to obtain preliminary data on the effect of TVT on visuo-cognitive function in Parkinson's, which will provide data to inform future clinical trials (i.e., sample size calculation etc.).

## Materials and methods

### Design

This pilot randomised controlled trial will compare a 4-week technological visuo-cognitive training intervention and a standard (non-technological) visuo-cognitive training intervention in PwP. Fig 1 outlines the SPIRIT schedule of enrolment, interventions and assessments. Participants will be screened for eligibility (-t1) and, once confirmed, will be randomised in a 1:1 allocation to one of the two visuo-cognitive intervention groups (technological or standard). They will then undergo baseline assessments (T1) before receiving 4 weeks of twice-weekly training sessions delivered by a physiotherapist in the participants' home. Following 4 weeks of visuo-cognitive training, post-intervention data collection will be conducted within one week of the final home visit (T2). Consent, data collection and interventions will all be

| | Enrolment | Allocation | Post-Allocation | |
|---|---|---|---|---|
| | -t1 | 0 | T1 | T2 |
| **TIMEPOINT** | | 0 | Baseline visit | Post 4-week intervention |
| **ENROLMENT:** | | | | |
| Eligibility screen | X | | | |
| Informed consent | X | | | |
| Allocation | | X | | |
| **INTERVENTIONS:** | | | | |
| Technological visuo-cognitive training | | | ●———————————————● | |
| Standard visuo-cognitive training | | | ●———————————————● | |
| **ASSESSMENTS:** | | | | |
| o  *Neuropsychological tests* | | | | |
| ▪ Geriatric depression scale (GDS-15) | | | X | - |
| ▪ Montreal cognitive assessment (MoCA) | | | X | - |
| ▪ Attention Computer Battery; simple and choice reaction time, digit vigilance | | | X | X |
| ▪ Forward digit span (seated) | | | X | X |
| ▪ Clock drawing/copying task (Royall's CLOX 1 &2) | | | X | X |
| ▪ Trail Making Test A and B | | | X | X |
| ▪ Benton's Judgement of Line Orientation Test | | | X | X |
| o  *Visual Sensory Functions* | | | | |
| ▪ LogMAR | | | X | X |
| ▪ Mars letter CS chart, Mars Percetrix™, USA | | | X | X |
| ▪ Senaptec Sensory Station | | | X | X |
| o  *Disease Specific Tests* | | | | |
| ▪ Hoehn & Yahr (H & Y) | | | X | - |
| ▪ The Unified Parkinson's Disease Rating Scale - III | | | X | X |
| ▪ The new FOG questionnaire (FOGQ) | | | X | X |
| ▪ Falls efficacy scale – International (FES-I) | | | X | X |
| ▪ Parkinson's Disease Questionnaire – PDQ 39 | | | X | X |
| ▪ Penn Parkinson's Daily Activities Questionnaire-15 | | | X | X |
| o  *Physical Performance Tests\** | | | | |
| ▪ Mini Best Test | | | X | X |
| ▪ Two-minute walk, single and dual task | | | X | X |
| ▪ The Fatigue Severity Scale | | | X | X |
| o  *Participation/Acceptability Measures* | | | | |
| ▪ Pittsburgh Rehabilitation Participation Scale | | | ●———————————————● | |
| ▪ Systems Usability Scale | | | ●———————————————● | |
| ▪ Semi-structured Interviews | | | | X |

**Fig 1. SPIRIT schedule of enrolment, interventions, and assessments.**

conducted in-person following national health recommendations for infection control during the COVID pandemic.

**Patient and public involvement.** The development and design of the visuo-training protocol and documentation for this study have involved consultation with six PwP through an online focus group conducted via Zoom in October 2020. Their opinions and preferences validated the relevance of the research concept and influenced the planned length and frequency of the study interventions.

**Study setting.** The setting for the study will be split between the clinical gait laboratory at Coach Lane Campus, Northumbria University, Newcastle upon Tyne (for assessment visits) and participant's home (for intervention sessions).

**Ethical approval and registration.** The design of this study conforms to the principles outlined in the Declaration of Helsinki and was approved by the South Central-Berkshire B Research Ethics Committee on 31 March 2021 (ref 21/SC/0042). Participation in the study will be voluntary and will require the written informed consent from each participant. Eligible patients will be informed about all relevant aspects of this study before commencing the study. This trial was listed on the ISRCTN registry with study ID ISRCTN46164906 on 21 April 2021.

**Participants: Sample size, recruitment and eligibility criteria.** Since this is a pilot study, a sample size calculation has not been performed [65, 66]. A convenience sample of forty community-living ambulatory older adults with mild-to-moderate Parkinson's disease will participate in this study in order to explore the practicalities of delivering the interventions in relation to the feasibility parameters outlined below [65]. Participants will be identified through attendance at Movement Disorders Clinics at Northumbria Healthcare NHS Foundation Trust and Gateshead Health NHS Foundation Trust, who have a combined total of over 1500 patients under follow-up. Additional participants will be recruited through the Parkinson's UK Research Support Network, Take Part Hub and local groups. As this study has been accepted onto the National Institute for Health Research Central Portfolio Management System (NIHR-CPMS ID 48327), the Dementias & Neurodegenerative Diseases Research Network (DeNDRoN) will also be involved for assistance with participant recruitment. The DeNDRoN network currently has several hundred PwP registered who have expressed their willingness to consider research participation. Links that have already been established through collaborative working and previous research projects conducted by study team will also be utilised. Recruitment is expected to continue until October 2022. This study conforms to the recommendations for reporting the results of pilot feasibility studies which are adopted from the CONSORT Statement [67].

**Inclusion criteria.** The inclusion criteria will be the following: participants aged >50 years with a clinical diagnosis of Parkinson's by a movement disorder specialist according to UK brain bank criteria (H&Y stage I-III) [68]. Participants must score a minimum of 21/30 on the Montreal Cognitive Assessment [69] to be able to follow instructions and undertake the training with support from the researcher. They must be living independently, able to walk and stand without support or assistance from another person, have adequate hearing/vision capabilities to allow participation in all aspects of study (if participant wears prescription glasses, they must be comfortable to remove these for short periods—up to 5 minutes at a time —in order to take part in activities whilst wearing strobe glasses) and stable medication for the previous 1 month and anticipated over a period of 6 months.

**Exclusion criteria.** The exclusion criteria will be: a history of epilepsy, seizures, migraines, severe motion sickness or sensitivity to light, psychiatric co-morbidity (e.g. major depressive disorder as determined by geriatric depression scale [70], clinical diagnosis of dementia or other severe cognitive impairment, history of stroke, traumatic brain injury, multiple sclerosis or neurological disorders other than Parkinson's disease, any acute lower back or lower

extremity pain, peripheral neuropathy, rheumatic and orthopaedic diseases, and unstable medical conditions including cardio-vascular instability in the past 6 months. If the participant is unable to comply with the testing protocol or currently participating in another interfering research project or undergoing any interfering therapy, they will not be recruited. Anyone experiencing Covid-19 symptoms will be managed as per latest government guidelines. Individuals who have *not* had the opportunity to receive both their Covid-19 vaccines will not be recruited although individuals may still be included in the study if they have been offered the vaccine but have declined due to personal circumstances.

## Randomisation and blinding

Once recruited, participants will be randomly allocated to two different groups. Randomisation will be performed via Research Randomizer (http://www.randomizer.org/) immediately after baseline assessment by a member of the research team. Participants will be randomised to receive the following distinct interventions: Group A–Technological visuo-cognitive training (*n = 20*) or Group B–Standard visuo-cognitive training (*n = 20*). It will not be possible to blind the physiotherapists delivering the interventions because of their active role in administering the visuo-cognitive training sessions. Video recordings of all post-intervention assessment proceedings will be undertaken to enable blinded members of the research team to review outcomes. Blinding will not be possible for participants due to the use of novel equipment in the technology arm of the trial, but they will be asked to refrain from mentioning the nature of the intervention they receive (technology versus standard) during their assessment in the Clinical Gait Laboratory. No participant-facing information (i.e., information sheets etc.) will refer to the potential superiority or effectiveness of TVT over the standard care intervention.

## Intervention

**Technological Visuo-cognitive Training (TVT).** Participants will receive a total of 8 sessions of TVT over a 4-week period. Each session will last up to 60 min of which 20 mins will be dedicated to a series of visuo-cognitive training drills using a mobile tablet device (see S2 File for a description of the drills) and a further 10–20 mins will be spent doing simple hand-eye co-ordination activities (e.g. throwing and catching) whilst wearing stroboscopic glasses (Fig 2). The remaining time will be factored in for rest periods as required. Please refer to S3 File for further details of the TVT intervention. Activities will be supervised at all times by a qualified physiotherapist from the research study team who will keep a running log of issues that may arise during the intervention period such as adverse events, changes in medication or any environmental factors that could affect participant engagement or performance.

**Standard care intervention.** Participants will receive a total of 8 sessions of standard (non-technological) visuo-cognitive training over a 4-week period. As there is currently no accepted "gold standard" visuo-cognitive training approach for use in PwP, the methodology for this study has been adapted from control conditions used in previous vision therapy trials and traditional therapy practice [61, 71]. Participants will undergo 10–20 minutes of throwing and catching drills (as per TVT intervention but without the stroboscopic glasses) and 20 minutes of visuo-cognitive training activities involving a variety of visuo-motor and perceptual tasks (primarily pen and paper). Refer to S3 File for a sample schedule of the standard care intervention activities.

## Participant engagement

The research team are confident that 40 participants can be recruited and retained by the trial based on our previous Parkinson's studies that have recruited similar numbers to exercise

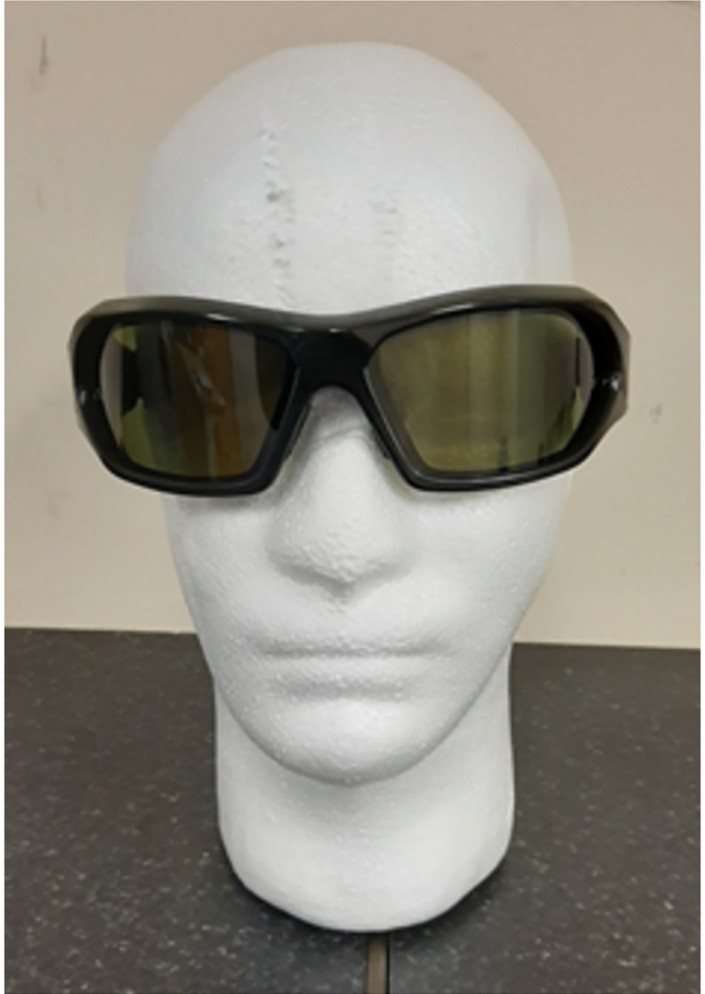

**Fig 2. Stroboscopic glasses.**

trials for 12-week exercise interventions [72]. Six PwP were consulted during a Public Participation Involvement (PPI) focus group event at the study planning phase. Feedback from the PPI session in relation to the length of the intervention and demands on participants was used to formulate the study design. Travel requirements and timescales will be explicitly discussed with participants at the outset to ensure they are fully aware of the expectations and time demands of the study, as well as potential impact on fatigue levels. Participants will receive a telephone call reminder ahead of each laboratory visit and scheduled home visit to ensure attendance. The study is expected to have a low attrition and a high compliance rate given that the intervention periods will be home-based with therapy supervision. In addition, the study has the added benefit of providing participants with the opportunity to receive a comprehensive evaluation of their visuo-cognitive abilities which is rarely offered as part of routine care.

## Outcomes and measurement

Once the necessary oral and written explanations have been provided to obtain their informed consent, a researcher will collect socio-demographic data from each participant regarding their age, gender, ethnicity, education level, falls history, activity levels and side dominance. In

addition, evaluation of cognition, visuomotor ability, disease severity, physical performance and quality of life will be performed by a physiotherapist from the research team. Assessments will be carried out with the participant in the ON medication phase (anti-parkinsonian medication taken one hour prior). The measures used for these assessments can be viewed in Fig 1.

A repeated measures design will be employed with assessments performed at the clinical gait laboratory during two separate sessions lasting between 2–3 hours. Participants will attend the lab for outcomes to be measured at the start (baseline-T1) and upon completion of the 4-weeks of visuo-cognitive training (T2).

## Feasibility

Feasibility will be explored in relation to the processes that are considered to be key to the success of the main study: feasibility of the intervention (stroboscopic glasses and mobile application) and feasibility of the study design and procedures [73]. Table 1 describes the feasibility outcomes for the study.

**Exploratory outcomes.** To collect preliminary data on the effects of TVT, we will assess change in visual attention from baseline to follow-up assessment (i.e., T1 to T2) using the Trail Making Test (TMT) [76]. The TMT is a well-known tool for investigating cognitive performance that depends on executive function, specifically, sustained attention and set-shifting [76, 77]. The TMT is a standardised assessment of the components of visuo-cognitive function that were being targeted by the TVT and that are known to be impaired in PwP, specifically visual attention and processing speed. Studies have shown that impairments in attentional capacity have a significant influence on balance and gait performance in PwP [16, 78, 79]. Not only is the TMT a good measure of overall cognitive function, it has also been shown to be a strong independent predictor of mobility impairment [80].

As a novel intervention in PwP, TVT has the potential to affect a variety of visual/motor/cognitive outcomes and we have included several secondary outcome measures to explore

**Table 1. Feasibility outcomes.**

| | Outcome | Description | Analysis |
|---|---|---|---|
| Feasibility of intervention | Safety | Adverse events (mild, moderate, or severe) caused by the intervention | Total number; compare differences in adverse events between the 2 arms. |
| | Amount | Duration of actual visuo-cognitive training undertaken during each visit | Compare average duration of visuo-cognitive training tolerated during each one hour visit [min] |
| | Compliance/ Adherence | Rate of scheduled and completed sessions | Total n sessions completed /total n sessions scheduled (%) |
| | Equipment | Function of mobile technology in participants' homes | Number of issues with equipment, such as break downs, shortage of equipment, internet access |
| | Acceptability | Useability | System Usability Scale [74] |
| | | | Conduct qualitative semi-structured interviews to explore issues around comfort and wearability of the glasses and implementation of the devices |
| Feasibility of study design and procedures | Acceptability | Motivation | Pittsburg Rehabilitation Participation Scale Scores [75] (compare mean scores between arms) |
| | Recruitment rates | Number of people approached who consent to take part. | Refusal rates; reasons for non-participation |
| | | | Are the eligibility criteria sufficient or too restrictive |
| | Participation rates | Do all eligible participants who agree to partake actually perform the training intervention? | Total number who undergo baseline assessment compared to total number who complete final testing. |
| | Assessment time scale | Can follow up data be collected within a week of completing the 4 weeks of training? | Number of participants whose follow-up data were collected within a week after the 4-week training period. |

these. Secondary outcomes will include changes in clinical outcomes, mobility (balance and gait), daily activities, quality of life, cognition, and falls-efficacy (see Fig 1 for list of outcomes and measures). Additionally, at baseline we will measure descriptive characteristics of the participants in terms of disease status, multimorbidity and mood.

Mobility and movement characteristics will be recorded via wearable sensors (e.g. Mobility Lab (version 2), Opals, APDM Inc., Portland, OR, USA; Invisible mobile eye-tracker, Pupil Labs, Germany) used during standing and walking under different conditions (e.g. single and dual task (forward digit span)). Spatiotemporal gait characteristics (e.g., gait speed (m/s), variability (coefficient of variation %), step/stride length (m), stride time (s), swing time (%), asymmetry, and step width (cm)) will be determined. Balance characteristics will include sway area ($mm^2$), sway path length (mm) and sway velocity (mm/s) [81]. Eye movement characteristics will include saccade frequency, amplitude, velocity and duration, and fixation duration [82, 83].

## Safety considerations

All measurements and interventions in relation to this study are non-invasive and place participants at no higher risk than those that normally may occur during sitting, standing, or walking. For some of the participants, particularly those who are not practicing any kind of physical exercise prior to the study, there is a slight possibility that they might feel some muscle soreness and fatigue after the interventions. To prevent excessive fatigue, participants will be encouraged to take breaks as needed throughout all study procedures. They will also be advised to consider the potential impact of fatigue on their Parkinson's symptoms when arranging their study visits.

It is possible that TVT may cause digital motion sickness (also known as *cyber sickness)*, a sensation similar to motion sickness which is caused by moving content on screens [84]. Although this phenomenon is generally related to highly immersive technology (such as virtual reality (VR) headsets) [85], the proposed study will deliver visuo-cognitive training at the most tolerable settings for individuals on the touch screen systems and strobe glasses to further minimise the risk of participants' experiencing these symptoms. A study by Kim et. al. (2017) looking at the use of VR by older adults and PwP during walking showed that they were able to use immersive technology without experiencing adverse effects [86] which provides reassurance that the non-immersive technological training component of the present study should not cause the participants any adverse effects.

The effects of using technology such as stroboscopic glasses as part of a visuo-cognitive training intervention in PwP is not yet known. Commonly recognised effects of prolonged exposure to strobe lighting such as that generated by flickering fluorescent lighting include headaches and eye fatigue, nausea and photosensitive epilepsy [87, 88]. Although participants will only be using the strobe glasses for up to 20 minutes at a time (including breaks), the eligibility criteria for this study has been designed to ensure that individuals who have a history of conditions that are known to be triggered by flashing/strobe lights (such as epilepsy or migraine) will be excluded from the study. Participants will also be given the opportunity to trial the strobe glasses prior to commencing procedures to ensure they can be tolerated. A qualified physiotherapist from the study team will always be with the participant whilst the technology is being used to monitor user-experience and ensure adequate rest periods are taken.

Any untoward medical occurrence, unintended disease or injury or any untoward clinical signs in participants whether or not related to the intervention will be recorded as an adverse event and managed according to the Health Research Authority (HRA) Guidance [89].

## Data analysis

Statistical analysis will be undertaken using SPSS version 25 or more recent versions (SPPS, Inc. an IBM company). A statistician (PO) blinded to group allocation will supervise the analysis. To check normality of data, results of the comparisons will be tabulated and box-plot graphs will be developed. Demographic characteristics and baseline data will be summarised using parametric or non-parametric descriptive statistics, as appropriate, for continuous or ordinal data, and percentages for categorical data. The participants or observations to be excluded, and the reason for their exclusion, will be documented prior to statistical analysis. Any exclusion documentation will be stored together with the remaining study documentation. A 95% confidence interval will be used, and statistical significance will be reported for all differences between groups at $p < 0.05$.

**Feasibility analysis.** Feasibility data will be presented as mean and standard deviation (normally distributed data), or median and nonparametric methods (non-parametric data). Differences in feasibility outcomes between the groups will be compared by two-tailed t-test (e.g., mean duration of visuo-cognitive training undertaken at each one-hour visit), or chi-square test (e.g. total number of differences in adverse events between arms, compliance/ adherence).

Feasibility outcomes will also be reported both descriptively and narratively in relation to usability and acceptability of the TVT package. A semi-structured interview will be performed at the end of the intervention period to explore participants' perception of the TVT intervention. Interviews will be carried out during the final home visit and will include a standardised introduction, followed by a selection of open ended questions and prompts to encourage further discussion and more specific answers in relation to their experiences of using the visual training equipment [90] (see S4 File for details of the interview schedule). Interviews will be audio-recorded and transcribed verbatim. A thematic analysis will be undertaken to identify patterns and key themes within the data [91].

**Efficacy analysis.** Efficacy data will be presented as mean and standard deviation (normally distributed data), or median and interquartile range (non-parametric data) for all quantitative measures. To determine the effect of the TVT package compared to standard care on our primary outcome of TMT (A and B), the difference between the baseline and post-intervention outcomes ($\Delta = T2-T1$) will be calculated. This will enable the comparison of the $\Delta$ values between the two groups (TVT and standard care) for each dependent variable. Additionally, repeated measure analysis of co-variance (ANCOVA) tests will be used to determine significance of change in our primary outcome of TMT (A and B) with the group (TVT vs standard care) as a between subject factor, while controlling for disease severity (UPDRS III), age and gender. Cohen's *d* effect sizes will be determined to evaluate the size of the effect.

## Management

The study will comply with the General Data Protection Regulation (GDPR) and Data Protection Act 2018, which require data to be de-identified as soon as it is practical to do so. All data will be entered into the database using unique study codes for each participant and will be securely stored on a password-protected computer database. Only members of the study team will have access to the data. Any significant protocol modifications during this study will be communicated to the trial registry. Data will be made available for other researchers following the end of the study at request to the principal investigator.

There will be no personal expenses for the participant during any stage of the study; all assessments and interventions will be free, and any travel expenses will be covered by the research budget if required. There will be no financial compensation for participation.

Participation in this study is completely voluntary and participants will be able to withdraw their participation in the study at any time, if desired, without any consequences related to their treatment or follow-up.

## Discussion

This will be the first pilot randomised study to compare TVT to standard care visuo-cognitive training in PwP, using state-of-the-art technology to assess and treat the visuo-cognitive systems. Visuo-cognitive function is a major area of decline in PwP. Despite increasing calls for further research into the area, treatment options for visuo-cognitive impairments in PwP remain limited to optimising dopaminergic therapy, traditional ophthalmological approaches and standard visuo-cognitive training (e.g. eye movement training, hand-eye co-ordination, pen and paper perceptual tasks etc.) [2, 11]. There is emerging evidence to suggest visuo-cognitive training may have a positive impact on visuo-cognitive function in neurological populations, but the effect in PwP remains unknown. Therefore, this study will examine the efficacy, feasibility (including usability), and overall safety of TVT for use in PwP. Below we outline the strengths and limitations of our study protocol.

Use of a home-based intervention performed by physiotherapists twice per week for a 4-week period enhances the generalisability of results to current clinical practices and provision-capacity within community rehabilitation services. An additional benefit of this study design is that all participants will receive a visuo-cognitive training intervention as part of their involvement in the study, whether it be with technology or without. As participant retention has been identified as an issue in a number of previous trials of physical rehabilitation in PwP and older adults [92, 93], it is anticipated that this will reduce the dropouts that have been associated with inactive control groups in other randomised controlled studies [94]. Furthermore, the semi-structured interviews will enable participants to share their experiences of using the strobe glasses and mobile application in their own homes. Usability has been shown to be an important factor to consider when analysing the feasibility of implementation and the usability of novel technologies for PwP [95, 96]. In an MDS commissioned review of mobile health technologies, Espay et. al. prioritised useability and compliance criteria as part of a protocol to evaluate new devices for use in PwP [97]. Use of the Systems Usability Scale and semi-structured interviews administered at the end of the intervention period will allow the study team to determine if TVT (combined stroboscopic eyewear and mobile application) is deemed acceptable and manageable for PwP.

One weakness of this study protocol that we acknowledge is the non-blinded assessment design that may expose the study to risks of bias in terms of selection and performance resulting in inflated treatment effects if the participants know their group allocation. Although blinding the participants to intervention assignment will not be possible in this study, participants will be reminded not to disclose their group status to the laboratory-based outcome assessor at any time. To reduce bias stemming from expectation, study participants will be blinded to the study hypotheses and the information provided in the participant information sheet will follow the principle of equipoise by declaring uncertainty about the superiority of one intervention arm over the other [98].

## Conclusion

This pilot study will contribute to our understanding of the feasibility and effects of TVT in PwP and provide preliminary data to support a larger, multi-centre RCT. Understanding the feasibility of TVT in PwP will help clinicians to develop tailored interventions using digital technologies for NMS that have received limited attention. If TVT involving the use of a

mobile application and stroboscopic glasses is demonstrated to be feasible and effective, it presents the possibility of a novel non-pharmacological rehabilitation strategy to improve visuo-cognitive function in Parkinson's disease.

## Supporting information

**S1 File. SPIRIT checklist.**
(TIF)

**S2 File. Description of drills.**
(TIF)

**S3 File. Interventions.**
(TIF)

**S4 File. Interview schedule.**
(TIF)

**S5 File. Protocol version 3.0 (Ethics approved).**
(DOCX)

## Acknowledgments

This study is based at the Physiotherapy Innovation Laboratory (Website: www.pi-lab.co.uk, Twitter: @Physio_In_Lab) and has been funded by a Northumbria University PhD studentship in collaboration with Senaptec Inc. (Beaverton, Oregon, USA) (PI: Dr Samuel Stuart). Dr Stuart is supported, in part, by a Parkinson's Foundation Post-doctoral Fellowship for Basic Scientists (PF-FBS-1898-18-21) and a Clinical Research Award (PF-CRA-2073).

## Author Contributions

**Conceptualization:** Julia Das, Rosie Morris, Gill Barry, Rodrigo Vitorio, Richard Walker, Samuel Stuart.

**Data curation:** Paul Oman.

**Formal analysis:** Julia Das, Paul Oman, Samuel Stuart.

**Funding acquisition:** Rosie Morris, Gill Barry, Richard Walker, Samuel Stuart.

**Investigation:** Julia Das, Rosie Morris, Gill Barry, Rodrigo Vitorio, Claire McDonald, Richard Walker, Samuel Stuart.

**Methodology:** Julia Das, Rosie Morris, Gill Barry, Rodrigo Vitorio, Paul Oman, Claire McDonald, Richard Walker, Samuel Stuart.

**Project administration:** Julia Das, Rosie Morris, Rodrigo Vitorio, Claire McDonald, Richard Walker, Samuel Stuart.

**Resources:** Samuel Stuart.

**Software:** Rodrigo Vitorio, Samuel Stuart.

**Supervision:** Rosie Morris, Gill Barry, Rodrigo Vitorio, Paul Oman, Richard Walker, Samuel Stuart.

**Validation:** Samuel Stuart.

**Writing – original draft:** Julia Das, Samuel Stuart.

**Writing – review & editing:** Julia Das, Rosie Morris, Gill Barry, Rodrigo Vitorio, Paul Oman, Claire McDonald, Richard Walker, Samuel Stuart.

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
