## [Decision Letter · Decision Letter 0]

12 Jul 2022

PONE-D-21-39411Exploring the effects of technological visuo-cognitive training in Parkinson’s: Study protocol for a pilot randomised controlled trial.PLOS ONE

Dear Dr. Stuart,

Thank you for submitting your manuscript to PLOS ONE. After careful consideration, we feel that it has merit but does not fully meet PLOS ONE’s publication criteria as it currently stands. Therefore, we invite you to submit a revised version of the manuscript that addresses the points raised during the review process.

Your manuscript has been assessed by two expert reviewers, whose comments are appended below. Reviewer 1 has highlighted concerns about several aspects of the study design, rationale and statistical analysis. Please ensure you respond to each point carefully in your response to reviewers document, and modify your manuscript accordingly.

We look forward to receiving your revised manuscript.

Kind regards,

Joseph Donlan

Editorial Office

PLOS ONE

Journal Requirements:

Reviewers' comments:

Reviewer's Responses to Questions

**Comments to the Author**

1. Does the manuscript provide a valid rationale for the proposed study, with clearly identified and justified research questions?

Reviewer #1: Partly

Reviewer #2: Yes

2. Is the protocol technically sound and planned in a manner that will lead to a meaningful outcome and allow testing the stated hypotheses?

Reviewer #1: Partly

Reviewer #2: Yes

3. Is the methodology feasible and described in sufficient detail to allow the work to be replicable?

Reviewer #1: Yes

Reviewer #2: Yes

4. Have the authors described where all data underlying the findings will be made available when the study is complete?

Reviewer #1: No

Reviewer #2: Yes

5. Is the manuscript presented in an intelligible fashion and written in standard English?

Reviewer #1: Yes

Reviewer #2: Yes

6. Review Comments to the Author

You may also provide optional suggestions and comments to authors that they might find helpful in planning their study.

Reviewer #1: Comments:

This small RCT addresses a gap in understanding and care and has the potential to contribute to the literature and body of knowledge in the field.

Some comments:

This is a feasibility pilot study to ‘provide preliminary data to support a larger, multi-centre randomised 46 controlled trial’. As such, the main outcomes (not primary, which should be reserved for a full scale efficacy/effectiveness trial) should be the feasibility/acceptability elements and the other efficacy outcomes (i.e. TMT) should be exploratory at best. Feasibility issues such as recruitment and attrition/retention should be included and at the forefront.

What is the rationale for the TMT as the main efficacy outcome?

Is this study addressing a priority area of research/clinical care stated by the Parkinson’s community? If so, it should be stated.

The rationale is ‘novel non-pharmacological rehabilitation strategy for prevention of falls in PwP’ but number and type of falls is not being measured. The overarching aim and outcomes need to be correctly aligned.

It is not clear why there is such an emphasis on the impact on visuo-cognitive function and falls? While this may certainly be an issue, the other functional impacts appear to be ignored – what about other ADLs, driving, using eating utensils, reading etc. The study is designed as a falls’ prevention program, yet the mechanism is broader, and the main outcome addresses visual attention (i.e. TMT) – this does not align correctly. The link between falls, visuo-cognition and visual attention need to be much better explained. The manuscript would benefit from a clearly thought through logic model.

The study compares two visuo-cognitive intervention groups (technological or standard); what is the rationale that one intervention will improve TMT more than the other? Surely the approaches differ only in mode of delivery/implementation rather than mechanism of action. Thus, the focus should be on adherence/acceptability/usability etc rather than a signal of change in cognitive outcome.

There are no cognition-based inclusion/exclusion criteria yet the main outcome, and indeed intervention, is visuo-cognition based. What about people with PD-MCI or parkinson’s plus syndromes?

What is the basis for the sample size? Has a power analysis been done?

What assessment will be done to ascertain probability of unblinding?

What is the rationale and evidence for the dose of the intervention (duration, frequency, number of weeks etc)?

Reviewer #2: This is a well-written, well-considered proposal for a pilot project. I think the protocol for persons with PD may be long but the authors have considered fatigue as a barrier. Perhaps specifying eye fatigue would be helpful as this is likely to occur in this population. I have no specific edits or recommendations beyond that and recommend accepting without revision.

7. PLOS authors have the option to publish the peer review history of their article (what does this mean?). If published, this will include your full peer review and any attached files.

Reviewer #1: No

Reviewer #2: **Yes: **Lisa M Muratori, PT, EdD

---

## [Author Response · Author response to Decision Letter 0]

29 Jul 2022

We would like to thank the editors and the reviewers for their hard work in reviewing our manuscript. We have added our reviewer responses within the file labelled "Response to Reviewers" and we hope that we have been able to respond satisfactorily to their comments and feedback.

---

## [Decision Letter · Decision Letter 1]

22 Sep 2022

Exploring the feasibility of technological visuo-cognitive training in Parkinson’s: Study protocol for a pilot randomised controlled trial.

PONE-D-21-39411R1

Dear Dr. Stuart,

We’re pleased to inform you that your manuscript has been judged scientifically suitable for publication and will be formally accepted for publication once it meets all outstanding technical requirements.

Kind regards,

Walid Kamal Abdelbasset, Ph.D.

Academic Editor

PLOS ONE

Additional Editor Comments (optional):

Reviewers' comments:

Reviewer's Responses to Questions

**Comments to the Author**

1. Does the manuscript provide a valid rationale for the proposed study, with clearly identified and justified research questions?

Reviewer #2: Yes

2. Is the protocol technically sound and planned in a manner that will lead to a meaningful outcome and allow testing the stated hypotheses?

Reviewer #2: Yes

3. Is the methodology feasible and described in sufficient detail to allow the work to be replicable?

Reviewer #2: Yes

4. Have the authors described where all data underlying the findings will be made available when the study is complete?

Reviewer #2: No

5. Is the manuscript presented in an intelligible fashion and written in standard English?

Reviewer #2: Yes

6. Review Comments to the Author

You may also provide optional suggestions and comments to authors that they might find helpful in planning their study.

Reviewer #2: No further comments. I accepted the manuscript previously and continue to believe the authors have a manuscript that is well-written and clear.

7. PLOS authors have the option to publish the peer review history of their article (what does this mean?). If published, this will include your full peer review and any attached files.

Reviewer #2: **Yes: **Lisa M Muratori

---

## [Editor Report · Acceptance letter]

28 Sep 2022

PONE-D-21-39411R1 

Exploring the feasibility of technological visuo-cognitive training in Parkinson’s: Study protocol for a pilot randomised controlled trial. 

Dear Dr. Stuart:

I'm pleased to inform you that your manuscript has been deemed suitable for publication in PLOS ONE. Congratulations! Your manuscript is now with our production department. 

Kind regards, 

on behalf of

Dr. Walid Kamal Abdelbasset 

Academic Editor

PLOS ONE